# Evaluation of Policies on Inappropriate Treatment of Dead Hogs from the Perspective of Loss Aversion

**DOI:** 10.3390/ijerph16162938

**Published:** 2019-08-15

**Authors:** Chenchen Yang, Jianhua Wang

**Affiliations:** 1School of Business, Jiangnan University, Wuxi 214122, China; 2Food Safety Research Base of Jiangsu Province, Jiangnan University, Wuxi 214122, China

**Keywords:** loss aversion, dead hogs, policy evaluation, hog production

## Abstract

Punishment policies on the inappropriate treatment of dead hogs play a key role in safeguarding public health and environmental protection. These policies aim to regulate the behavior of farmers and promote the development of sustainable agriculture. Farmers’ evaluation of a policy can be used to measure its effectiveness, and loss aversion is a factor that has been little studied. This study surveyed 404 hog farmers in China, and analyzed the factors that influenced their evaluation of the penalties for the inappropriate treatment of dead hogs during 2016 and 2017. We used three indicators for the evaluation of the penalties: the degree of necessity, implementation, and effectiveness. Special attention was paid to farmers’ aversion to financial penalties and police detention time, which was elicited using economic experiments. The results show that farmers are more likely to be averse to police detention time than financial penalties, and suggest that the level of each indicator needs to be increased. The results from an ordered Probit model show that there are both similarities and differences between the formation paths of the three indicators. An aversion to financial penalties will help to improve the degree of implementation. An aversion to police detention time will lead to a negative trend in the degree of effectiveness. An in-depth analysis of the factors that influence farmers’ evaluation of policies to punish inappropriate treatment of dead hogs may provide a basis for the design of government policies to improve environmental protection performance.

## 1. Introduction

Inappropriate treatment of dead hogs pollutes the environment and threatens human health. To ensure that hog products are safe and of high quality, China’s government has been attempting to reduce the inappropriate treatment of dead hogs. For example, the government has promulgated a series of policies and regulations on hog production, including the Animal Epidemic Prevention Law in 2007, the Interpretation of Several Issues Concerning the Application of Laws in Handling Criminal Cases Hazardous in 2013, and the Opinions on Establishing a Mechanism for Harmless Treatment of Dead Livestock and Poultry in 2014. These policies prohibit farmers from inappropriately handling dead hogs. The relevant provisions of the Law of the People’s Republic of China on Animal Epidemic Prevention state that persons who dispose of dead animals (including dead hogs) for an unknown cause will be punished. The relevant agencies shall order the offender to conduct harmless treatment of animals, require the offender to bear the handling costs, and impose a fine on the offender of not more than 3000 yuan. If the circumstances are serious, they shall be addressed based on the specific situation. The acts of producing or selling pork products from sick animals are deemed to be illegal and persons who perform them will receive punishment. Persons who sell dead hogs no longer only receive a fine, and those who cause a serious food poisoning incident or other serious incident involving a foodborne disease shall be sentenced to a fixed-term imprisonment of not more than three years. If the consequences are serious, the offender will be sentenced to more than seven years in prison or life imprisonment. The Opinions on Establishing a Mechanism for the Harmless Treatment of Dead Livestock and Poultry stipulates that in cases where animals (including dead hogs) are slaughtered, sold, or have an unknown cause of death, the relevant department must order the perpetrator to take remedial measures, confiscate the products, and impose a fine of more than five times the value of the goods. The penalties for the inappropriate handling of dead hogs in the existing law and regulations are primarily fines, with detention as a supplement. Therefore, this paper considered both fines and detention.

What is the effect of a punishment policy on the implementation process? How does a farmer evaluate the policy? Follow-up investigations on a policy’s implementation are one of the means through which to ensure its effectiveness. Understanding farmers’ evaluation of a policy on the inappropriate treatment of dead hogs is a premise of and the basis for its effective implementation and continuous improvement. Therefore, analyzing the formation process of a policy’s evaluation based on punishment policy evaluation indicators is of theoretical and practical importance. Investigating the role played by farmers’ characteristics, animal breeding characteristics and breeding conditions, and other factors in policy evaluation may provide a specific direction for improving farmers’ policy evaluations and the optimization of policies. With the deepening of prospect theory in the field of policy evaluation, psychological factors have gradually become important indicators in policy evaluation research. The value function in prospect theory consists of a risk preference coefficient, a loss aversion measure, and a nonlinear probability weighting coefficient [1]. The loss aversion measure refers to the ratio of the absolute value of the negative utility produced by losing one unit of wealth to the absolute value of the positive utility of giving one unit of wealth when a person makes a future decision; that is to say, losses can have a greater psychological effect than returns [2,3]. Therefore, losses are often more difficult to accept than the same amount of earnings [4]. Provisions on losses in public policy can fully amplify the loss that people suffer when they behave inappropriately, and lead people to generate an aversion to loss to avoid the occurrence of inappropriate behavior. 

The objective of this study was to understand the characteristics of farmers and their evaluation of a punishment policy on the inappropriate handling of dead hogs through an in-depth investigation. In other countries, food safety regulation is more stringent. In the United States, the listing of genetically modified (GM) foods is subject to review by three departments. Several years are needed before any GM food enters the market, and an outlay of $10 to $15 million is required to collect the data to complete the approval process, which considerably reduces the number of incidents involving unsafe food. In Germany, food safety issues can be traced back to the country of origin. Once food safety problems are discovered, the German government quickly isolates pig and poultry farms and forces the slaughter of affected livestock. The Japanese government comprehensively intervenes in and supervises food safety, and the compensation awarded by the court in food safety cases is generally reasonable. The government has established a liability insurance fund to provide compensation to victims in product liability cases. Based on the perspective of an aversion to loss, this paper explored the formation mechanism of farmers’ evaluation of policies on the inappropriate treatment of dead hogs. Finally, suggestions were provided for improving the level of farmers’ satisfaction of these policies and using farmers’ aversion to loss to construct efficient and reasonable policies.

The rest of this paper is structured as follows. Section 2 presents a literature review. Section 3 describes the data source and presents descriptive statistics. Section 4 outlines the experimental design; the results are provided in Section 5. Our conclusions and discuss policy implications are presented in the final section.

## 2. Literature Review

The research on policies has mainly focused on the status quo of policies, the transmission mechanism of policy evaluation, and the formulation of policy evaluation indicators. With regard to policy status quo, farmers say that the government’s policy on inappropriate treatment of dead hogs is enforceable, and the degree of influence of supervision and punishment policies on farmers’ behavior is obvious [5]. However, some scholars have reported that the punishment policies on the inappropriate handling of dead hogs are insufficient to deter farmers from this behavior. In the long term, non-standard treatment of dead hogs will occur. Due to the arbitrariness of a farmer’s inappropriate handling of dead hogs, the probability of being discovered and punished is low, which limits the policy’s effectiveness [6]. With regard to the transmission mechanism of policy evaluation, Crabbé stated that it is crucial to identify the path to a policy’s implementation in a policy evaluation study [7]. As agriculture is the industry with the longest development history, the government has issued many policies to ensure its sustainable development. Clarifying the problems in the process of a policy’s implementation by evaluating agricultural policies is a research topic in the field of agriculture. For example, Tang et al. assessed the level of performance of support policies through provincial and agricultural commodity levels and suggested that China’s agricultural product policy needs to be further improved [8]. Specifically, an important way of improving the level of performance of an agricultural product policy is to improve the technical efficiency of agricultural products. Moser et al. used framed experiments and games to provide respondents with different stimulations (rewards and penalties) to obtain their preferences from a producer’s perspective [9]. The results showed that the probability of the implemented policy and the direction of the stimulus are the key indicators that influence the policy’s effectiveness, and a high return fertilizer policy with a low probability is the most motivating policy. In a study of the sustainable development of food safety in Ireland, Lynch showed that the refinement, expansion, and transformation of sustainable development indicators can promote the evaluation and development of agricultural policies [10]. In an analysis of the effect of an agricultural policy’s implementation, breeding characteristics, cultivation period, breeding conditions [11], education level, and income level are generally the influencing factors [12,13]. 

In our exploration of the indicators for measuring a policy’s evaluation and the development of evaluation mechanisms, this work found that not many studies have been conducted on the policy indicators for managing dead hogs. Therefore, the work drew on the research conducted in other fields and divided the policy evaluation index into three indicators: the degree of necessity, the degree of implementation, and the degree of effectiveness. This is similar to the conclusions of Barbiroli on evaluation indicators for China’s urban green transformation policy [14]. The author pointed out that the three core factors that ultimately identify and refine the evaluation index system of urban green transformation are policy attributes, the implementation process, and the implementation effects. Zhang used policy formulation, policy implementation, and policy results as the evaluation criteria to measure the overall implementation effect in Hubei province in an evaluation of the effectiveness of high-tech industrial policies [15].

This work incorporated psychological factors into the variable system to expand our understanding of the formation mechanism of a policy’s evaluation, and the relationship between the degree of aversion to loss of policy evaluators and a policy’s evaluation to analyze the impact of loss aversion was selected. In our study of relevant policies, our work did not find many studies that considered psychological factors, and few penalties were found regarding policies on the inappropriate handling of dead hogs. Drawing on policy research in other fields, Huang and Du evaluated China’s housing security policy from the perspective of public satisfaction [16]. The results showed that when residents’ level of trust and fairness is higher, their satisfaction with housing security is higher. Xia et al. introduced the free patriarchal system into the design of public policy when discussing the implementation of public policy on land use control [17], which involves considering the characteristics of people’s aversion to loss to enhance the acceptance of public policy and optimize existing policies. The results of their analysis show that an increase in the use of compensation as a method of control, a reduction in law-abiding losses, an increase in the use of penalties for illegal transfers, and an increase in the prohibitions in the law can comprehensively improve a policy’s effectiveness. Kibeta et al. found that risk aversion promoted the recognition of GAP (Good Agricultural Practices) certification among bean growers, and loss avoidance was not conducive to the popularity of the certification [18].

After reviewing the literature, two gaps were found to exist in the research. The first is that the research has not included the degree of aversion to loss in the analysis of the factors that influence a policy’s evaluation, and ignores the guiding role that social psychological factors play in an evaluation of a policy on the inappropriate treatment of dead hogs. The existing research on farmers’ policy evaluation and the transmission mechanism of policy evaluation does not include many psychological factors. Therefore, analyzing the effects of psychological factors of farmers on their evaluation of a policy and identifying the mechanism of the implicit factors is crucial to optimizing the policy’s design and improving farmers’ recognition of and satisfaction with the policy. Therefore, the contribution of this paper is a policy evaluation that includes a loss aversion factor. The second gap is that the existing data on psychological factors were mostly obtained directly using a Likert scale or a risk equivalent. These data have a large subjective component and are consequently of low value. A few studies on farmer policies introduced experimental design methods, such as answer selection. The experiments by Tanaka et al. [19] and Holt and Laury [20] were used to provide the basis for our research methods.

## 3. Theoretical and Research Framework

### 3.1. Theoretical Framework

The theoretical framework was constructed based on prospect theory (PT) and assumed that farmers are bounded rational decision-makers [21]. Kahneman and Tversky proposed PT and found that individuals are less sensitive to returns [3]. When facing a loss, the economic behavior that individuals exhibit is an appetite for risk, whereas they are considered risk averse in the case of gains. In particular, loss aversion is the degree of an individual’s evasion of loss, and, in cumulative PT, the value function is based on the return and loss, that is, the degree of positive and negative deviation from the decision point of the decision-maker. The degree of negative deviation is expressed by the loss. The loss is weighted by a loss aversion measure, and the weight that is assigned to the loss is larger, so that the benefit is treated differently from the loss. To explain the existence of loss aversion, some scholars have used evolutionary theory, which suggests that people prefer more conservative and cautious behavior because aversion to loss in decision-making can improve their survival and reproduction success rate; so, to improve survival and evolution, people prefer loss aversion [22]. In the field of behavioral economics, the multiple price list (MPL) method is a common economic experimental method for measuring risk preference or loss aversion. This method originated from research that Binswanger conducted on Indian farmers [23]. More recently, domestic scholars have used this method to measure farmers’ risk aversion and study its impact on agricultural production factors [24] and pesticide application behavior [25]. The MPL method is easy to implement, its subject matter is easy to understand, and it more realistically represents risk appetite. Based on PT, Tanaka et al. constructed an MPL experimental design that was used to measure the time preference coefficient [19], the risk preference coefficient, and a loss aversion measure. The authors also designed a lottery decision experiment that consisted of three parts and 35 pairs, including risky returns, risky losses, and combinations of different probabilities of these gains or losses. This work used the MPL Tanaka et al. [19] and Holt and Laury’s [20] methods to measure loss aversion in pig farmers.

### 3.2. Research Framework

Based on the literature review and theoretical framework, the developed research framework is shown in Figure 1.

## 4. Data Source and Sample Characteristics

### 4.1. Data Source

The data used in this study were obtained from a questionnaire survey of hog farmers in 12 provinces in China. The survey was administered using a layered design and random sampling technique and was conducted by Jiangnan University from January to March 2018. Due to the agglomeration and difference in scale of hog production among provinces, the intensity of hog breeding was divided into three categories: large, medium, and small. This paper used the number of live pigs in each province in 2016 as the standard for dividing the large, medium, and small categories. In the survey’s first stage, 12 provinces were selected according to the geographical distribution and hog production scale of the sample provinces. The survey area’s distribution is shown in Figure 2. In the survey’s second stage, two to three sample areas (counties) were selected in provinces with a large amount of hog production, and one to two sample areas (counties) were selected in provinces with a medium or small volume. In the survey’s third stage, this paper randomly selected 15–20 farmers for investigation in each county. Since the investigators are local residents, we can obtain information on all pig farmers through them and their related personnel as well as local village committees. On this basis, we randomly selected farmers. To ensure the validity of the survey and the reliability of the data, experts in the relevant field were invited to review and optimize the questionnaire and a pre-survey was conducted in Anhui province and Jiangsu province. Finally, a total of 450 questionnaires were distributed and 404 valid questionnaires were received after data cleaning.

Face-to-face interviews were conducted by well-trained interviewers who were hired from local universities in each province. Before the interview, the farmers received questionnaires issued by the investigators. Our survey was structured, but the investigators needed to provide assistance during the process of filling out the questionnaire to solve the confusion of farmers. The survey collected information on household-level and farm-level characteristics (e.g., age, sex, education, and hog livestock status) and external environmental characteristics for the loss aversion measure.

### 4.2. Variable Description and Sample Characteristics

Table 1 presents the definitions of and descriptive statistics for the variables that were used in this study. Individual characteristics, family characteristics, and animal breeding characteristics were included as control variables that affect the treatment of dead hogs in the system of factors influencing a punishment policy’s evaluation. External conditions and environmental variables are expressed in terms of farmers’ production conditions and the organizational environment. What needs special explanation is that “External environmental characteristics” mean the support and help that the government and society can provide during the breeding process. As an individual’s aversion to loss is considerably affected by long-term growth in their environment, the socio-economic differences between the surveyed areas are significant. Therefore, regional dummy variables (Reg) were added to control for regional fixed effects. According to the data from many Internet companies on cities, the comprehensive strength of the 338 prefecture-level cities in China in 2017 was ranked Second-tier cities and cities above the second tier in the survey area were assigned a value of 1, and cities below the second-tier were assigned a value of 0.

The results of Table 1 show that 70.3% of the respondents were male, with few female farmers, the average age was 49.33 years old, the average education level was 7.34 years, and 96% of the farmers were married. Regarding family characteristics, the 21%–40% proportion of income from pig farming was the largest, and the number of people in the other four sections was roughly the same. The internal differentiation of pig farmers in China was relatively large. In terms of the proportion of people engaging in hog production labor, the number of people in the range 21%–40% was the highest (38.1%), and the number of people in the range of 81%–100% only accounted for 1.5%. Most farmers were engaged in hog breeding work, and family members constituted the main labor force. The average time spent as a breeder was 12.72 years, and most of the farmers had rich experience in pig farming. Regarding the breeding scale variable, the majority of people fell into the interval between 0–50 and 51–500 pigs, accounting for 53.5% and 41.8% of the respondents, respectively. Only 4.7% of the respondents had a breeding scale of more than 500 pigs, and a considerable number of farmers conducted free-range pig farming. A total of 84.9% of the surveyed areas were cities below the second tier, with a general level of development. A total of 44.8% of the farmers indicated that there were no diseased pigs in the breeding area. Of the respondents, 68.8% indicated that they had not joined a cooperative, and the level of popularity of pig breeding cooperatives was low.

### 4.3. Ordered Probit Model

A farmer’s evaluation of a punishment policy on the inappropriate treatment of dead hogs is a multi-ordered variable. The ordered Probit model is suitable for managing cases where the dependent variable is an ordered multi-class discrete variable. In the dependent variable that was selected in this study, the degree of necessity indicator was divided according to the five-point Likert scale as follows: Y1 = 1, which means that the policy is totally unnecessary; Y2 = 2, which means that the policy is unnecessary; Y3 = 3, which means that the policy is generally necessary; Y4 = 4, which means that the policy is necessary; and Y5 = 5, which means that the policy is very necessary. The degree of effectiveness and the degree of implementation indicators were similarly divided according to the five-point Likert scale.

The general form of the model can be expressed as follows:(1)Y∗=β0+β1Gender+β2Age+β3Edu+β4Ms+β5Income+β6Lab+β7Exp+β8Scale+β9Reg+β10Disposal+β11Cooperative+β12LA1/LA2+ξ,
where is a latent variable, is a residual term that is subject to a normal distribution with a variance of σ, $\beta_{0}$, $\beta_{1}$, $\beta_{2}$, …, $\beta_{12}$ are the coefficients to be estimated. Let Y denote the farmers’ evaluation of the punishment policy. The greater the value, the more positive the farmers’ evaluation of the policy. Assuming there are tangent points $k_{1},k_{2},k_{3},$ the relationship between Y and $Y^{*}$ can be expressed as follows:(2)$$Y=\left\{\begin{array}{l} 1\;\mathrm{if}\;Y^{*}\leq k_{1}\\ 2\;\mathrm{if}\;k_{1}<Y^{*}\leq k_{2}\\ 3\;\mathrm{if}\;k_{2}<Y^{*}\leq k_{3}\\ 4\;\mathrm{if}\;k_{3}<Y^{*}\leq k_{4}\\ 5\;\mathrm{if}\;Y^{*}>k_{4} \end{array}\right..$$

Writing Equation (1) in its vector form: this paper can derive:(3)$$\begin{array}{l} \mathrm{Pr}(Y=1)=\mathrm{Pr}(Y^{*}\leq k_{1})=\mathrm{Pr}(X\beta +\xi \leq k_{1})\\ =\mathrm{Pr}(\xi \leq k_{1}-X\beta )=\Phi (k_{1}-X\beta ) \end{array}.$$

As such, this paper can obtain:(4)$$\mathrm{Pr}(Y=2)=\Phi (k_{2}-X\beta )-\Phi (k_{1}-X\beta ),$$
(5)$$\mathrm{Pr}(Y=3)=\Phi (k_{3}^{}-X\beta )-\Phi (k_{2}-X\beta ),$$
(6)$$\mathrm{Pr}(Y=4)=\Phi (k_{4}-X\beta )-\Phi (k_{3}-X\beta ),$$
(7)$$\mathrm{Pr}(Y=5)=1-\Phi (k_{4}-X\beta ),$$
where Φ(•) represents the cumulative distribution function of the general normal distribution. Unlike in the general least squares estimation method, the explanatory variables in the multi-ordered Probit model describe a probability problem, whose solution can be estimated by the maximum likelihood method [26].

## 5. Experimental Design

To obtain more realistic micro-scale data on each respondent farmer’s degree of aversion to loss, this paper measured the degree of aversion to loss using an experimental economics method. The authors Kemel et al., using a loss aversion measure [27], found a difference between the degree of aversion to a loss of personal freedom and the degree of aversion to a loss of money. When a public policy involves detention and fines, an individual will exhibit a different degree of aversion to loss. With respect to detention, people have a higher degree of aversion to loss [27]. Therefore, two items, detention and fines, were set to obtain and compared different loss aversion measures. Based on the MPL proposed by Tanaka et al. [20] and Holt and Laury [21], this paper inputs the punishments of fines and detention for the inappropriate treatment of dead hogs into the MPL, which enabled the farmers to understand and perceive the experimental design more realistically to improve the experiment’s authenticity and effectiveness. In the experiment’s first stage, to check whether the respondents truly understood the meaning of the questions that were being asked, at the beginning of the experiment, the investigators introduced the rules of the experiment to the respondents and helped the respondents to follow them.

The investigators asked the respondents: “If the local government department was to impose penalties on pig production farmers who randomly dispose of or sell dead hogs, which option would you choose in the following two situations?” Table 2 lists the options.

According to each farmer’s response, the investigators recorded the farmer’s choice between the two situations. Since the first case has a certain probability of generating a fine of 1000 yuan but there is also the possibility of not being fined, farmers with a lower degree of loss aversion will choose this option. In contrast, although the second case will result in a fine (500 yuan) that is lower than that in the first, it is an inevitable event, so those with higher degree of loss aversion will prefer this option to avoid high fines. If the respondent was able to make a choice and provide a reasonable explanation, then this paper considered the responder to understand the question thoroughly and conducted a formal experiment. When the respondent indicated that they did not understand the question or arbitrarily answered it, then the investigator was required to explain the question in more detail until the respondent truly understood, and then conducted the formal experiment.

In the experiment’s second stage, after the respondents were familiarized with the experiment’s rules, the investigators provided two sets of 20 multiple-choice questions. In the MPL’s first group, 10 groups of fines for the inappropriate handling of dead hogs were set as the multiple-choice questions for the study sites (Table 3). In the MPL’s second group, 10 groups of detention periods for the inappropriate treatment of dead hogs were set as the multiple-choice questions (Table 4). In both sets of experimental protocols, the probability of being fined was 50%, which was designed to exclude farmers from being affected by probability weighting. The first option in the MPL’s first group was 100% probability of being fined 50 yuan or 100–500 yuan, and the first group of the MPL’s second group was 100% probability of being detained for 1 or 2 h. At 10 h, the same period of detention and the amount of fines corresponded one-to-one. According to the principle of high risk and high return, the first case in each MPL group is a low risk option, and the second case is a high risk option, allowing respondents to choose between the two schemes. In the experiment’s third stage, based on the final choices of the respondents, this paper was able to provide the option to jump from the first case to the second case. According to each farmer’s actual selection, this paper calculated their loss aversion measure: LA1 = (the selection of the first case or the second case in the fine question)/10, LA2 = (the option from the first case to the second case in the detention question)/10. If a loss aversion measure is 1, the respondent’s loss aversion is high. If a loss aversion measure is 0, the respondent’s loss aversion is low. The investigators asked the respondents: “The local government department imposes fines on pig farmers who randomly dispose of or sell dead hogs. Will you choose the first or the second case?” Table 3 shows the options. Investigators were required to emphasize that the investigation would not affect the respondent’s farm’s breeding qualifications and would not affect their reputation and information was collected for research purposes only. Given farmers’ perception of the punishment policy, and to avoid unrealistic answers due to farmers’ resistance, the questionnaire indicated that “this paper only knows your opinion about the fine and will not report you”. This paper required the investigators to make this clear to the farmers during the investigation.

The investigators asked the respondents: “The local government department will detain pig farmers who randomly dispose of dead hogs. Would you choose the first case or the second case?” Table 4 shows the options. Investigators were required to emphasize that the investigation would not affect the respondent’s farm’s breeding qualifications and would not affect their reputation.

Figure 3 shows the frequency distribution of farmers’ aversion index for fines and detention losses. The penalty and detention loss aversion index is the highest in 0.5–0.6. The number of farmers with a loss aversion index greater than 0.5 is higher than the number of farmers with a loss aversion index less than 0.5. This result shows that most farmers had a higher degree of aversion to loss, whether facing fines or detention. In the first experimental protocol, the average loss aversion index of the farmers was 0.59. In the second experimental protocol, the average loss aversion index of the farmers was 0.61. These results show that the loss aversion index for detention was slightly larger than that for fines.

## 6. Empirical Analysis

### 6.1. Current Status of Farmers’ Evaluation of Punishment Policies for the Inappropriate Treatment of Dead Hogs

In this study, farmers’ evaluation of the punishment policy for the inappropriate treatment of dead hogs was divided into three indicators according to the five-point Likert scale—An evaluation of the degree of necessity (PN), the degree of implementation (PI), and the degree of effectiveness (PE) of the punishment policy. This paper then collected data on the farmers’ evaluation of the punishment policy in the survey area. These data are shown in Table 5.

The data shown in Table 5 represent the current status of the farmers’ evaluation of the punishment policy. Firstly, in the evaluation of the degree of necessity of the punishment policy (PN), the “necessary” and “very necessary” options accounted for 35.6% of the farmers, and only 10.7% of the farmers think that the punishment policy is unnecessary. That is, most farmers had a positive evaluation of the policy in terms of its degree of necessity. For the degree of implementation (PI), 51.2% of the farmers provided a score of three or below, indicating that most farmers believed that the degree of implementation of the punishment policy is low. Finally, regarding the evaluation of the degree of effectiveness (PE) of the punishment policy, 59.9% of the farmers believed that the punishment policy is mostly effective in regulating the behavior of the farmers, and 40.1% of the farmers provided scores of three or below. The above analysis shows that problems remain with the current status of farmers’ evaluation of the punishment policy, and the overall evaluation is not positive, especially in terms of implementation and effectiveness.

### 6.2. Results and Discussion

In this study, this paper investigated the factors that influence farmers’ evaluation of a punishment policy for the inappropriate treatment of dead hogs. The dependent variable in Models I and II is the evaluation of the degree of necessity of the punishment policy. Model I adds a loss aversion measure of a pecuniary penalty and Model II adds a loss aversion measure of detention time. The dependent variable in Models III and IV is the evaluation of the degree of implementation of the punishment policy. Model III adds a loss aversion measure of a pecuniary penalty and Model IV adds a loss aversion measure of detention time. The dependent variable in Models V and VI is the evaluation of the degree of effectiveness of the punishment policy. Model V adds a loss aversion measure of a pecuniary penalty and Model VI adds a loss aversion measure of detention time. The analysis was performed using Stata version 15.1 (STATA, North Carolina, USA). Table 6, Table 7, Table 8, Table 9, Table 10 and Table 11 shows the results. This paper estimates the effect on the probability of choosing each category (e.g., the effect on the probability of choosing “Unnecessary = 1” to “Unnecessary = 5” in the necessary for the punishment policy), separately, and we use “Margin (1)” to “Margin (5)” to express the probability.

The pseudo coefficient of determination (*R*^2^) values in Table 6 suggest that our estimations have a good degree of fit across the models. This paper first analyzed the effect of loss aversion. In Model II, the respondents’ aversion to detention time has a positive effect on the evaluation of the degree of policy implementation at the 1% level. That means that the higher the loss aversion, the better the evaluation of the degree of necessity. In Model III, the respondents’ aversion to the amount of the fine in the penalty policy has a positive effect on the evaluation of the degree of policy implementation at the 5% level: the greater the coefficient of the loss aversion at the level of the monetary sum, the smaller the amount of speculative psychology, and the better the evaluation of the degree of implementation. The loss aversion measure of detention time did not have a significant effect on the evaluation of the degree of implementation. From the perspective of loss aversion, when a farmer receives a fine as punishment for the inappropriate handling of dead hogs and this produces resistance, the better the evaluation of the degree of the policy’s implementation. However, farmers’ degree of aversion to loss in terms of detention does not significantly affect their evaluation of the degree of the policy’s implementation. The reason for this may be that the amount of the fine in the punishment policy can be quantified, is easy to implement, and is more relevant to the interests of most farmers. Therefore, the degree of aversion to a monetary loss is more related to the evaluation of the degree of the policy’s implementation. The particulars of a detention policy may be vague and difficult to implement. Policy-makers can use farmers’ aversion to the amount of the fine and its impact on the level of implementation to develop policies that are more restrictive and better regulate the behavior of farmers. Notably, the loss aversion measure is different in Model VI. In Model VI, after adding an abandonment coefficient of the time limit for the detention policy, the loss aversion measure was found to be negatively affected by the degree of effectiveness of the punishment policy at the 10% significance level: the stronger the farmers’ aversion to a loss of personal freedom caused by a punishment policy, the lower their evaluation of the effect of the punishment policy on the disposal of dead hogs. The possible reason for this is that in real life, the government penalties that farmers face in the study area or in the surrounding areas are mostly punishments involving money. Only when there is a major epidemic or a food poisoning incident will the persons involved be detained. This paper concludes that the detention time clause in the punishment policy has a poor effect on farmers’ inappropriate handling of dead hogs. Therefore, the greater the aversion coefficient of the farmers’ evaluation of the amount of the fine, the more the farmers believe that the current punishment policy is effectively regulating their treatment of dead hogs. The greater the aversion coefficient of the farmers’ loss with respect to detention time, the more the farmers believe that the current punishment policy’s detention time is not able to effectively regulate their treatment of dead hogs. Accordingly, policy-makers should provide a more detailed description of the detention duration when creating policies to penalize farmers for inappropriate treatment of dead hogs.

Regarding the six models, the animal breeding scale was found to have a positive impact on the evaluation of degree of necessity, the degree of implementation, and the degree of effectiveness of the punishment policy. The reasons for the impact of scale and labor were found to be similar. The more energy and financial resources that farmers invest in the breeding of hogs, the higher the degree of specialization in hog breeding. Farmers prefer the government to impose penalties. The policy regulates the hog breeding market, creates a good market environment, and prevents the occurrence of inappropriate disposal of dead hogs. In the external environment, disposal points for dead hogs were found to have a positive impact on the degree of necessity, the degree of implementation, and degree of effectiveness of the punishment policy at the 1% and 5% levels of significance. The construction of equipment to dispose of dead hogs is the measure that the government uses to regulate the disposal of dead hogs. The availability of disposal equipment makes farmers feel that the government is paying attention to problems associated with the disposal of dead hogs and raising awareness about the harmless disposal of dead hogs, thus improving the farmers’ evaluation of the degree of necessity of a punishment policy. The existence of disposal equipment for dead hogs improves the efficiency of the disposal of dead hogs, and the government can transmit a message of safe disposal of disease-carrying hogs. Additionally, government-created disposal sites for dead hogs have a higher degree of regulation of the disposal of dead hogs, and are more strict in the implementation of their policies, which may increase farmers’ evaluation of the degree of implementation and effectiveness of punishment policies.

Finally, the unique characteristics of farmers were found to be factors that influenced the different indicators of punishment policy evaluation. In Models I and II, the income level of the farmers had a positive impact on the perception of degree of necessity of the punishment policy at the 1% significance level. The higher the income level, the higher the degree of necessity. The higher the household’s income, the more likely farmers are to protect their property and to require policies to protect them. The labor was found to have a positive impact on the degree of implementation and degree of effectiveness at the 1% significance level. The larger the workforce that farmers allocate to hog breeding, the more they hope that the hog market functions in a virtuous circle. Therefore, the higher their evaluation of the indicators of the punishment policy, the better their evaluation of the degree of implementation and the degree of effectiveness. In Models I–IV, breeding time was negatively affected by the degree of implementation at the 1% significance level. The descriptive statistics show that the farmers’ evaluation of the implementation degree of the punishment policy is poor, indicating that the current punishment policy has not been well-implemented. Farmers with more years of experience in hog breeding would have been aware of this phenomenon for a longer period, so their evaluation of the degree of implementation would be worse. In Models I, II, V, and VI, at the external environment level, farmers who had joined a hog breeding cooperative were found to have a better evaluation of the degree of effectiveness. Farmers that participate in hog breeding cooperatives can be exposed to more normative management and systematic training, have a higher familiarity with and pay more attention to government policies, and better evaluate the effectiveness of the punishment policy.

The marginal effect from the Probit model enabled us to identify different degrees of change for each evaluation indicator. For example, in Model I, when the sex variable is changed from female to male, Margin (2) is 0.003 and Margin (4) is −0.002, which indicates that the evaluation of the degree of necessity first rises and then falls. As the trends are all different and there are many variables, this paper will not repeat them all here.

### 6.3. Policy Recommendations

Based on our results, this paper provides the following policy recommendations. First, the human capital structure of hog farmers should be optimized, and a high-quality farmer team should be built. In mass media, reports of highly educated individuals returning to the countryside to plant or breed are increasing, and highly educated farmers rely on higher education levels and large-scale farming to achieve greater economic benefits. Therefore, both the media and the government ought to provide more positive publicity to encourage college students to return to their hometowns to improve the quality and level of the entire group of farmers. Second, moderate-scale breeding should be promoted and the degree of specialization of farmers should be increased. If large-scale farms (households) invest more in farming and are more dependent on the generated income, they will face greater barriers to performing inappropriate actions. Publicity and training to improve the level of professional farming can encourage farmers to focus on advanced technology for handling diseased dead hogs rather than on dealing with penalties for violations. Third, more equipment for the disposal of dead hogs should be constructed and farmers should be encouraged to participate in hog breeding cooperatives. Strengthening the supervision of the construction of disposal sites for dead hogs, implementing the government’s preferred policies for the disposal of dead hogs, and providing more convenient channels for the disposal of dead hogs are additional recommendations. A hog breeding cooperative should be constructed and implemented to create a standardized breeding environment for farmers and farmers should be encouraged to participate actively. Fourth, more stringent regulations and laws should be formulated to combat misconduct and strengthen the government’s enforcement of inappropriate behavior. The penalties for fines and detention should be clearer, the costs associated with violations should be increased, the idea of farmers’ taking risks with respect to violations needs to be dispelled, and legal means should be used to enhance the compliance of farmers with laws. The handling of dead hogs must be standardized. Fifth, the “boost” approach to normative behavior should be used. The purpose of boosts in behavioral economics is to influence people’s choices and make them profitable on the basis of decision-makers’ judgments. According to this, after an incident involving the inappropriate disposal of dead hogs and a legal ruling, the local government may require the hog farmers to report the progress of the incident and publicize it via SMS. A boost uses the psychology of people’s aversion to loss to guide them to make correct choices in an unenforced way, and to subtly strengthen their awareness of legal norms. Sixth, economic methods should be used to determine the optimal amount of a fine and detention time. A method for specifying the amount of a fine and the time of detention through formulas is worth developing. A potential formula is penalty strength = the loss caused by misconduct/the probability of misconduct being discovered.

## 7. Conclusions

In this study, this paper investigated the loss aversion of hog production farmers using an experimental economics method, and examined the determinants of loss aversion using an ordered Probit model. This paper used questionnaire survey data collected from 404 hog production farmers from 12 provinces in China in 2018. The results show that farmers are more averse to detention time than to fines. Kemel et al. suggested that in public policy, individuals are more averse to personal security threats than losses from fines [27]. These two conclusions are similar. Regarding the three indicators of a policy’s evaluation, farmers’ evaluation of the effectiveness is low, and their evaluation of the degree of implementation level is the lowest. Breeding scale and disposal points for dead hogs were both found to be associated with each of the three indicators to a certain extent. The remaining variables had different effects on different dependent variables. The aversion coefficient of the time of detention in the punishment policy had a positive effect on the evaluation of the degree of necessity of the policy. The aversion coefficient of the amount of the fine in the punishment policy had a positive effect on the evaluation of the degree of implementation, and the aversion coefficient of the time of detention in the punishment policy had a negative effect on the evaluation of the degree of effectiveness of the policy. With respect to the problem of endogeneity, this paper selected as many control variables as possible that affect both the explanatory variables and the explained variables to address any endogenous problems, such as missing variables [28,29].

## Figures and Tables

**Figure 1 ijerph-16-02938-f001:**
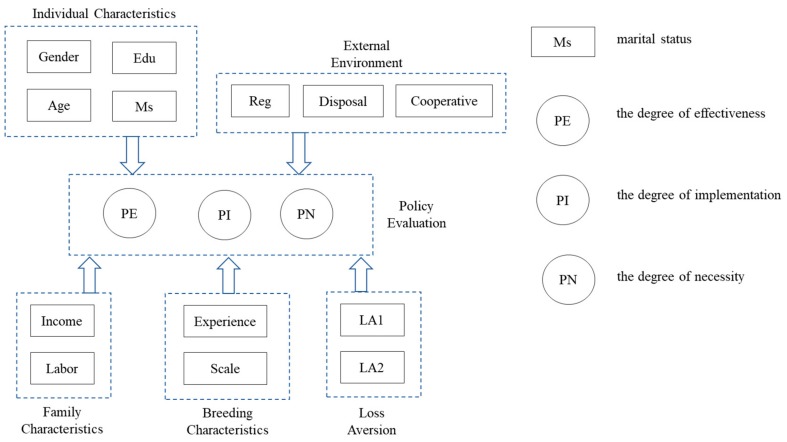
Research framework for factors that influence a punishment policy’s evaluation.

**Figure 2 ijerph-16-02938-f002:**
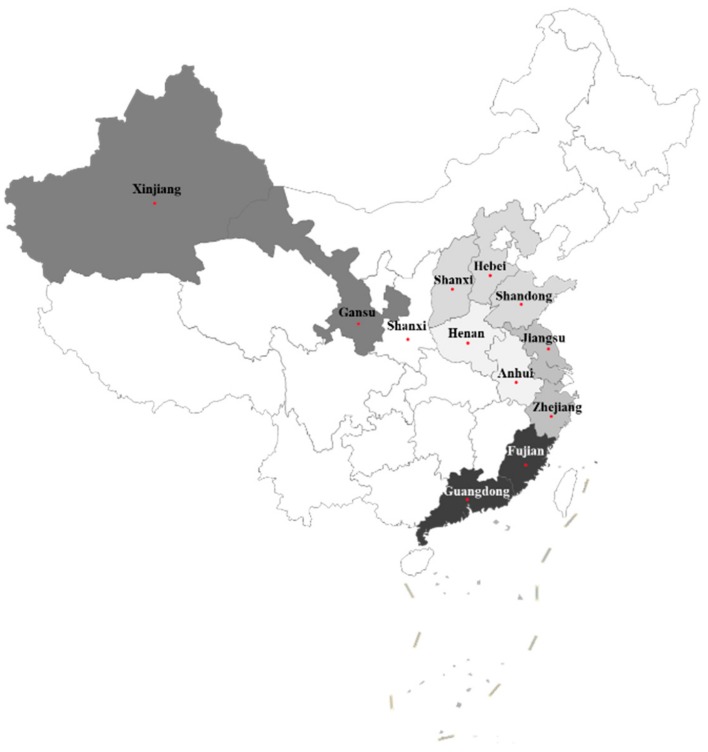
A map of the survey area. Note: The color change from dark to light indicates a larger to smaller number of surveyed people, respectively.

**Figure 3 ijerph-16-02938-f003:**
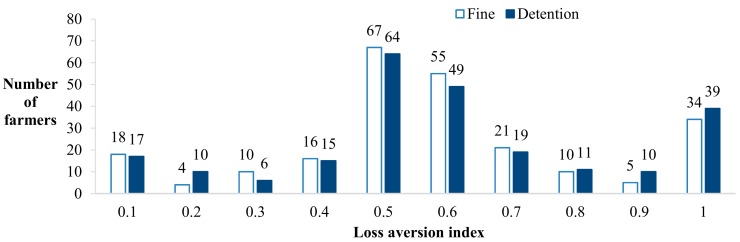
Frequency distribution of the aversion coefficient of loss to the amount of the fine and the detention time of farmers.

**Table 1 ijerph-16-02938-t001:** Variable selection and feature analysis.

Variable		Variable Assignment	Frequency (Mean)	Frequency (Standard Deviation)
**Individual Characteristics (IC)**	Sex	Female = 0	120	29.7%
		Male = 1	284	70.3%
	Age	Years	(49.33)	(8.68)
	Education	Schooling years	(7.34)	(4.39)
	Marital Status	Unmarried = 0	16	4%
		Married = 1	388	96%
**Family Characteristics** **(FC)**	Income	0%–20% = 1	73	18.1%
		21%–40% = 2	110	27.2%
		41%–60% = 3	86	21.3%
		61%–80% = 4	72	17.8%
		81%–100% = 5	63	15.6%
	Labor	0%–20% = 1	102	25.2%
		21%–40% = 2	154	38.1%
		41%–60% = 3	98	24.3%
		61%–80% = 4	44	10.9%
		81%–100% = 5	6	1.5%
**Breeding** **Characteristics** **(BC)**	Experience	Numerical Value	(12.72)	(9.22)
	Scale	0–51 = 1	216	53.5%
		51–500 = 2	169	41.8%
		501–3000 = 3	14	3.5%
		3001–10,000 = 4	4	1.0%
		10,001 and above = 5	1	0.2%
**Dummy** **Variable (DV)**	Reg	Cities below the second tier = 0	343	84.9%
		Second-tier cities and above = 1	61	15.1%
**External Environment (EE)**	Disposal	No = 0	181	44.8%
		Yes = 1	223	55.2%
	Cooperative	No = 0	278	68.8%
		Yes = 1	126	31.2%

Note: Reg = regional dummy variables.

**Table 2 ijerph-16-02938-t002:** Options in the experimental exercise.

Question Number	The First Case	The Second Case
1	50% may be found and a fine 1000 yuan, and 50% may not be found	100% found and a fine of 500 yuan

**Table 3 ijerph-16-02938-t003:** Punishment policy penalty amount options (Group 1).

Question Number	The First Case	The Second Case
1	100% found and a fine of 50 yuan	50% may be found and a fine of 500 yuan
2	100% found and a fine of 100 yuan	50% may be found and a fine of 500 yuan
3	100% found and a fine of 150 yuan	50% may be found and a fine of 500 yuan
4	100% found and a fine of 200 yuan	50% may be found and a fine of 500 yuan
5	100% found and a fine of 250 yuan	50% may be found and a fine of 500 yuan
6	100% found and a fine of 300 yuan	50% may be found and a fine of 500 yuan
7	100% found and a fine of 350 yuan	50% may be found and a fine of 500 yuan
8	100% found and a fine of 400 yuan	50% may be found and a fine of 500 yuan
9	100% found and a fine of 450 yuan	50% may be found and a fine of 500 yuan
10	100% found and a fine of 500 yuan	50% may be found and a fine of 500 yuan

**Table 4 ijerph-16-02938-t004:** Punishment policy detention time options (Group 2).

Question Number	The First Case	The Second Case
1	100% found and detention for 1 h	50% may be found and detention for 10 h
2	100% found and detention for 2 h	50% may be found and detention for 10 h
3	100% found and detention for 3 h	50% may be found and detention for 10 h
4	100% found and detention for 4 h	50% may be found and detention for 10 h
5	100% found and detention for 5 h	50% may be found and detention for 10 h
6	100% found and detention for 6 h	50% may be found and detention for 10 h
7	100% found and detention for 7 h	50% may be found and detention for 10 h
8	100% found and detention for 8 h	50% may be found and detention for 10 h
9	100% found and detention for 9 h	50% may be found and detention for 10 h
10	100% found and detention for 10 h	50% may be found and detention for 10 h

**Table 5 ijerph-16-02938-t005:** Frequency distribution of farmers’ evaluation of punishment policies for the inappropriate treatment of dead hogs.

Variable		Variable Assignment	Frequency	Rate of Recurrence
**Policy evaluation**	How necessary do you think the punishment policy is for the inappropriate treatment of dead hogs? (PN)	Totally unnecessary = 1	12	3.00%
		Unnecessary = 2	31	7.70%
		General = 3	73	18.10%
		Necessary = 4	144	35.60%
		Very necessary = 5	144	35.60%
	How well do you think the local government is implementing the punishment policy for the inappropriate treatment of dead hogs? (PI)	Not executed at all = 1	27	6.70%
		Basically not Executed = 2	72	17.80%
		General = 3	108	26.70%
		Mostly executed = 4	109	27.00%
		Fully executed = 5	88	21.80%
	How effective do you think the current policy is for the inappropriate treatment of dead hogs? (PE)	Totally ineffective = 1	21	5.20%
		No effect = 2	38	9.40%
		General = 3	103	25.50%
		Mostly effective = 4	139	34.40%
		Very effective = 5	103	25.50%

**Table 6 ijerph-16-02938-t006:** Estimation results of the formation mechanism of policy evaluation (Model I).

Variable	Model I					
	Coefficients (DV = PN)	Margin (1)	Margin (2)	Margin (3)	Margin (4)	Margin (5)
Sex	−0.010 (0.118)	0.002	0.003	0.001	−0.002	−0.004
Age	0.005 (0.007)	−0.001	−0.001	−0.000	−0.000	0.001
Education	0.016 (0.276)	0.001	0.001	–0.000	–0.001	–0.002
Marital Status	0.015 (0.019)	–0.002	–0.002	–0.000	0.001	0.003
Income	0.156 *** (0.054)	–0.024	–0.028	–0.006	0.017	0.041
Labor	0.048 (0.062)	–0.006	–0.007	–0.001	0.004	0.010
Experience	–0.016 ** (0.007)	0.002	0.003	0.001	–0.002	–0.004
Scale	0.204 * (0.106)	–0.037	–0.042	–0.009	0.026	0.062
Reg	0.197 (0.173)	–0.026	–0.030	–0.007	0.019	0.044
Disposal	0.391 *** (0.117)	–0.060	–0.069	–0.015	0.043	0.101
Cooperative	0.241 ** (0.116)	–0.036	–0.041	–0.009	0.025	0.060
LA1	0.001 (0.021)	–0.009	–0.010	–0.002	0.006	0.015
LA2						
Observations	404					
Log likelihood	–203.751					
Prob > chi^2^	0.000					
Pseudo *R*^2^	0.159					

Note: *, **, and *** denote significance at the statistical levels of 10%, 5%, and 1%, respectively. The numbers in parentheses are standard errors. DV means dependent variable.

**Table 7 ijerph-16-02938-t007:** Estimation results of the formation mechanism of policy evaluation (Model II).

Variable	Model II					
	Coefficients (DV = PN)	Margin (1)	Margin (2)	Margin (3)	Margin (4)	Margin (5)
Sex	–0.015 (0.118)	0.002	0.003	0.001	–0.002	–0.004
Age	0.004 (0.007)	–0.001	–0.001	0.000	0.000	0.001
Education	0.008 (0.275)	0.001	0.001	0.000	–0.001	–0.002
Marital Status	0.012 (0.019)	–0.002	–0.002	0.000	0.001	0.003
Income	0.164 *** (0.053)	–0.024	–0.028	–0.006	0.017	0.041
Labor	0.039 (0.062)	–0.006	–0.007	–0.001	0.004	0.010
Experience	–0.016 ** (0.007)	0.002	0.003	0.001	–0.002	–0.004
Scale	0.251 ** (0.107)	–0.037	–0.042	–0.009	0.026	0.062
Reg	0.177 (0.173)	–0.026	–0.030	–0.007	0.019	0.044
Disposal	0.408 *** (0.117)	–0.060	–0.069	–0.015	0.043	0.101
Cooperative	0.244 ** (0.116)	–0.036	–0.041	–0.009	0.025	0.060
LA1						
LA2	0.060 *** (0.023)	–0.009	–0.010	–0.002	0.006	0.015
Observations	404					
Log likelihood	–203.982					
Prob > chi^2^	0.000					
Pseudo *R*^2^	0.158					

Note: *, **, and *** indicate significance at the statistical levels of 10%, 5%, and 1%, respectively. The numbers in parentheses are standard errors.

**Table 8 ijerph-16-02938-t008:** Estimation results on the formation mechanism of policy evaluation (Model III).

Variable	Model III					
	Coefficients (DV = PI)	Margin (1)	Margin (2)	Margin (3)	Margin (4)	Margin (5)
Sex	–0.161 (0.119)	0.039	0.013	–0.011	–0.023	–0.018
Age	0.011 (0.007)	–0.003	–0.001	0.001	0.002	0.001
Education	0.005 (0.019)	–0.021	–0.007	0.006	0.012	0.010
Marital Status	0.086 (0.288)	–0.001	0.000	0.000	0.001	0.001
Income	0.050 (0.054)	–0.012	–0.004	0.004	0.007	0.006
Labor	0.255 *** (0.064)	–0.062	–0.021	0.018	0.037	0.029
Experience	–0.018 *** (0.007)	0.004	0.002	–0.001	–0.003	–0.002
Scale	0.296 *** (0.108)	–0.072	–0.025	0.021	0.042	0.034
Reg	0.237 (0.173)	–0.058	–0.020	0.017	0.034	0.027
Disposal	0.738 *** (0.118)	–0.180	–0.062	0.052	0.106	0.084
Cooperative	0.119 (0.118)	–0.029	–0.010	0.008	0.017	0.014
LA1	0.044 ** (0.021)	–0.011	–0.004	0.003	0.006	0.005
LA2						
Observations	404					
Log likelihood	–550.397					
Prob > chi^2^	0.000					
Pseudo *R*^2^	0.107					

Note: *, **, and *** denote significance at the statistical levels of 10%, 5%, and 1%, respectively. The numbers in parentheses are standard errors.

**Table 9 ijerph-16-02938-t009:** Estimation results on the formation mechanism of policy evaluation (Model IV).

Variable	Model IV					
	Coefficients (DV = PI)	Margin (1)	Margin (2)	Margin (3)	Margin (4)	Margin (5)
Sex	–0.163 (0.119)	0.040	0.014	–0.012	–0.024	–0.019
Age	0.010 (0.007)	–0.002	–0.001	0.001	0.001	0.001
Education	0.004 (0.019)	–0.006	–0.002	0.002	0.004	0.003
Marital Status	0.025 (0.286)	–0.001	0.000	0.000	0.001	0.000
Income	0.026 (0.053)	–0.006	–0.002	0.002	0.004	0.003
Labor	0.256 *** (0.064)	–0.063	–0.022	0.018	0.037	0.029
Experience	–0.019 *** (0.007)	0.005	0.002	–0.001	–0.003	–0.002
Scale	0.333 *** (0.109)	–0.081	–0.029	0.024	0.048	0.038
Reg	0.213 (0.172)	–0.052	–0.018	0.015	0.031	0.024
Disposal	0.761 *** (0.118)	–0.186	–0.065	0.054	0.110	0.087
Cooperative	0.127 (0.118)	–0.031	–0.011	0.009	0.018	0.015
LA1						
LA2	0.016 (0.023)	–0.004	–0.001	0.001	0.002	0.002
Observations	404					
Log likelihood	–552.298					
Prob > chi^2^	0.000					
Pseudo *R*^2^	0.104					

Note: *, **, and *** denote significance at the statistical levels of 10%, 5%, and 1%, respectively. The numbers in parentheses are standard errors.

**Table 10 ijerph-16-02938-t010:** Estimation results on the formation mechanism of policy evaluation (Model V).

Variable	Model V					
	Coefficients (DV = PE)	Margin (1)	Margin (2)	Margin (3)	Margin (4)	Margin (5)
Sex	–0.096 (0.119)	0.026	0.006	–0.013	–0.010	–0.009
Age	0.006 (0.007)	–0.002	0.000	0.001	0.001	0.001
Education	0.022 (0.019)	0.034	0.008	–0.017	–0.013	–0.012
Marital Status	–0.125 (0.282)	–0.006	–0.001	0.003	0.002	0.002
Income	0.002 (0.054)	0.000	0.000	0.000	0.000	0.000
Labor	0.253 *** (0.064)	–0.068	–0.016	0.034	0.027	0.024
Experience	–0.009 (0.007)	0.002	0.001	–0.001	–0.001	–0.001
Scale	0.559 *** (0.112)	–0.151	–0.035	0.075	0.059	0.052
Reg	0.169 (0.174)	–0.046	–0.011	0.023	0.018	0.016
Disposal	0.289 ** (0.118)	–0.078	–0.018	0.039	0.031	0.027
Cooperative	0.259 ** (0.119)	–0.070	–0.016	0.035	0.027	0.024
LA1	0.022 (0.022)	–0.006	–0.001	0.003	0.002	0.002
LA2						
Observations	404					
Log likelihood	–527.046					
Prob > chi^2^	0.000					
Pseudo *R*^2^	0.094					

Note: *, **, and *** denote significance at the statistical levels of 10%, 5%, and 1%, respectively. The numbers in parentheses are standard errors.

**Table 11 ijerph-16-02938-t011:** Estimation results on the formation mechanism of policy evaluation (Model VI).

Variable	Model VI					
	Coefficients (DV = PE)	Margin (1)	Margin (2)	Margin (3)	Margin (4)	Margin (5)
Sex	–0.091 (0.120)	0.024	0.006	–0.012	–0.009	–0.008
Age	0.006 (0.007)	–0.002	0.000	0.001	0.001	0.001
Education	0.024 (0.019)	0.037	0.009	−0.018	−0.014	−0.013
Marital Status	−0.137 (0.282)	−0.006	−0.001	0.003	0.002	0.002
Income	−0.017 (0.053)	0.004	0.001	−0.002	−0.002	−0.002
Labor	0.263 *** (0.064)	−0.071	−0.017	0.035	0.028	0.025
Experience	−0.010 (0.007)	0.003	0.001	−0.001	−0.001	−0.001
Scale	0.549 *** (0.113)	−0.147	−0.035	0.073	0.057	0.051
Reg	0.179 (0.174)	−0.048	−0.011	0.024	0.019	0.017
Disposal	0.292 ** (0.118)	−0.078	−0.018	0.039	0.031	0.027
Cooperative	0. 264 ** (0.119)	−0.071	−0.017	0.035	0.028	0.025
LA1						
LA2	−0.042 * (0.023)	0.011	0.003	−0.006	−0.004	−0.004
Observations	404					
Log likelihood	−525.860					
Prob > chi^2^	0.000					
Pseudo *R*^2^	0.096					

Note: *, **, and *** denote significance at the statistical levels of 10%, 5%, and 1%, respectively. The numbers in parentheses are standard errors.

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
