# Peer review of "Evaluation of Policies on Inappropriate Treatment of Dead Hogs from the Perspective of Loss Aversion"

_ijerph, 2019, doi:10.3390/ijerph16162938_

Round 1

Reviewer 1 Report

Title: Using Loss Aversion to Support Sustainable Environmental Development in Policy-Designing

Comments: Major Revise

This study analyzes the determinants influencing the evaluation of the penalties for the irregular treatment of dead hogs. Based on the perspective of loss aversion, the formation mechanism of farmers' policy evaluation of irregular treatment of dead hogs is explored. The major comments are listed as follows:

1.     Title. The authors should revise the title to be closer to your research topic. Environmental Development is in your title and key words, however, I can hardly see more information about Environmental Development in your innovation point, indicator selection and study results. Please explain how to reflect the Environmental Development in the policy-designing and your study. Besides, dead hogs is the object of the study and it should be reflect in the title.

2.     Abstract. It should reflect the research gap.

3.     Introduction. IJERPH is an international journal. This paper fails to engage with the wider readership of the journal. You list lots of policies related to China. What is the international situation? What’s the difference between China and other countries in the policy? You should present clearly and this can help the paper to attract more international readers.

4.     Scientific originality/novelty: The novelty/originality should be clearly justified that the manuscript contains sufficient contributions to the new body of knowledge from the international perspective.

5.     Literature review. Research gap should be presented in the Introduction. Research gap should be reorganized, and the existing state is unacceptable.

Line 123, you said “this paper draws on the research in other fields”. What do you mean the “other fields”?

Line 126-132, you present the study divides the policy evaluation index into three indicators, which is similar to the study of Barbiroli (2011) and Zhang Lan (2015). So what’s your contribution in the indicator selection?

6.     The research methods used in the paper should be clearly described.

7.     Theoretical framework. Please combine your theory with your research content and show your logical thinking and mechanism through a Figure.

8.     Data. “The intensity of 204 hog breeding is divided into three categories: large, medium and small.” What is the basis of division?

Figure 1, what’s the different color mean? Please add the legend for Fig.1.

9.     Results: It should be modified to Results and Discussions.

Table 2. Why the unit in the first case is yuan and in the second case in bucks.

Line 327-328, the author present implementation degree (PI) in this part, while in Line 125, it was implementation level, please be consistent.

The probit model should be presented in the Model part, rather than the Result.

10.  Conclusions. Please shorten and refine the first paragraph in the conclusions.

Others:

1.     The language should be improved. Grammar errors are very common in this paper. e.g. Line 74-76, Line 94-97, Line 133-135, Line 231-234, Line 381…..

It is recommended that it should be edited by whom is familiar with the scientific language before submission.

2.     There are many repetitive sentences in the text. Line 122 and 123, Line 133 and 135. The authors should check the full text carefully and avoid similar mistakes.

Author Response

Response to Reviewer 1

Comments: Major Revise

This study analyzes the determinants influencing the evaluation of the penalties for the irregular treatment of dead hogs. Based on the perspective of loss aversion, the formation mechanism of farmers' policy evaluation of irregular treatment of dead hogs is explored. The major comments are listed as follows:

Point 1. Title. The authors should revise the title to be closer to your research topic. Environmental Development is in your title and key words, however, I can hardly see more information about Environmental Development in your innovation point, indicator selection and study results. Please explain how to reflect the Environmental Development in the policy-designing and your study. Besides, dead hogs is the object of the study and it should be reflect in the title.

Response 1. In response to your suggestion, we have now revised the title of the paper as: “Study on the Policy Evaluation of Irregular Treatment of Dead Hogs from the Perspective of Environmental Concern”, and we changed the key word “environment development” to “environmental concern”. We use the term “environmental concern” to mean that standardizing treatment of dead hogs is based on environmental protection considerations. At present, the phenomenon of littering and throwing dead hogs poses a serious threat to the environment. The policy of establishing dead hogs disposal equipment mentioned in the policy design and other policies regulating dead hogs are designed to protect the ecological environment from damage.

Point 2 Abstract. It should reflect the research gap.

Response 2. In response to your suggestion, we have now added the sentence  “Loss aversion is a factor that has not been studied too much” into the abstract to reflect the research gap more Clearly .

Point 3.  Introduction. IJERPH is an international journal. This paper fails to engage with the wider readership of the journal. You list lots of policies related to China. What is the international situation? What’s the difference between China and other countries in the policy? You should present clearly and this can help the paper to attract more international readers.

Response 3. We have used several sentences to list the international situation in the introduction. Please refer to the introduction.

Point 4. Scientific originality/novelty: The novelty/originality should be clearly justified that the manuscript contains sufficient contributions to the new body of knowledge from the international perspective.

Response 4. In order to clearly justify scientific originality, we added the sentence into the last paragraph in the introduction. Please refer to the introduction.

Point 5. Literature review. Research gap should be presented in the Introduction. Research gap should be reorganized, and the existing state is unacceptable.

Line 123, you said “this paper draws on the research in other fields”. What do you mean the “other fields”?

Line 126-132, you present the study divides the policy evaluation index into three indicators, which is similar to the study of Barbiroli (2011) and Zhang Lan (2015). So what’s your contribution in the indicator selection?

Response 5. We put the research gap in the research review because we believe that the research gap should be reasonable after domestic and foreign literature analysis, and we have reorganized the gap.

Line 123, other fields indicate other policies that remove the evaluation indicators for dead hogs policies, such as China's urban green transformation policy.

Line 126-132,our contribution is to broaden the boundaries of the use of evaluation indicators, and our research content is different from the subject of the paper we borrowed.

Point 6. The research methods used in the paper should be clearly described.

Response 6. We have clearly described the methods used in the paper. Applying experimental economics to loss aversion measures, using ordered Probit model in model selection.

Point 7. Theoretical framework. Please combine your theory with your research content and show your logical thinking and mechanism through a Figure.

Response 7. The theory explains the existence of loss aversion, we added our logical thinking and mechanism in Figure 1.

Point 8. Data. “The intensity of 204 hog breeding is divided into three categories: large, medium and small.” What is the basis of division?

Figure 1, what’s the different color mean? Please add the legend for Fig.1.

Response 8. Because the number of farmers in different provinces is different, we use the indicator of the amount of production to determine the approximate number of people surveyed. The standard of classification is the number of pig production. The main step is to sort all the provinces first. According to the number of columns, the number of provinces in the three categories is equal. Then consider the geographical distribution to finalize the survey province.

We have added the legend for Fig.2(now).

Point 9. Results: It should be modified to Results and Discussions.

Table 2. Why the unit in the first case is yuan and in the second case in bucks.

Line 327-328, the author present implementation degree (PI) in this part, while in Line 125, it was implementation level, please be consistent.

The probit model should be presented in the Model part, rather than the Result.

Response 9. In response to your suggestion, we have now changed Results into Results and Discussions, bucks into yuan, the evaluation of the implementation level into the evaluation of the implementation degree.

The probit mode has been presented in the Model part

Point 10. Conclusions. Please shorten and refine the first paragraph in the conclusions.

Response 10. In response to your suggestion, we have now shortened and refined the first paragraph in the conclusions.

Others:

Point 1. The language should be improved. Grammar errors are very common in this paper. e.g. Line 74-76, Line 94-97, Line 133-135, Line 231-234, Line 381…..

It is recommended that it should be edited by whom is familiar with the scientific language before submission.

Response 1. We have revised the language by whom is familiar with the scientific language before submission.

Point 2.  There are many repetitive sentences in the text. Line 122 and 123, Line 133 and 135. The authors should check the full text carefully and avoid similar mistakes.

Response 2. We have checked the full text carefully and avoided similar mistakes.

Reviewer 2 Report

This article investigates how farmers evaluate financial penalties and police detention for the irregular treatment of dead hogs, as well as how specific factors influence their evaluation. The evaluation of punishment policies is divided into three different pillars: the necessity, the degree of implementation and the effectiveness. In addition, based on an economic experiment, the manuscript has the potential to add to the literature on how these components of policy evaluation interact with farmer’s loss aversion. Results indicate that farmers have a positive evaluation on the necessity degree of punishment policies even though they believe that policies are not executed well. Furthermore, loss aversion has some effect on the above mentioned pillars and farmers are more averse to detention than financial penalties. Below follows a list of comments and suggestions for the authors on how the manuscript could be improved.

Broad comments

At some parts of the manuscript, it is extremely difficult for the reader to follow the meaning because of the incoherent English. There are many grammatical incorrect sentences and it would be advisable for the authors to rewrite some parts or take the help of professional editing.

I would suggest the authors to reconsider the title of the paper and use a more declarative and informative title that doesn’t exaggerate the findings.

In introduction and literature review sections, the relevance of the research is not well described, the key hypothesis is difficult to be deducted from these parts and should be elaborated on.

Please re-evaluate the use of the phrase “among them”. It is used too often and sometimes it is used in positions that is not necessary (e.g. lines 107, 354). 

Specific comments

1.       The second paragraph in introduction confuses the reader. In lines 80-83, be more clear on explaining the loss aversion coefficient. In line 84, you might want to talk about “earnings” rather than “income”.

2.       In line 220, did you actually collect data for aquaculture production? That mean fishes… You use the same word in lines 118 and 486. Please check the meaning.

3.       Please elaborate more on the variables description section. In lines 248-249, you describe the variable “culture scale”. What is this? Do you mean the farm scale? And the counting 0-50, 50-500… do you refer to the number of pigs per livestock farm? Please explain. Furthermore, the scale here is not correct. It should be 0-50, 51-500, 501- 3000 etc, otherwise you have overlaps in your calculations.

4.       In lines 251-252, please explain why the percentage of the cities below the second line is so bigger than the percentage of the second tier cities and above? Does this disproportion affect your results and how? Please elaborate more on this.

5.       In the loss aversion question, there is a discrepancy between the training question and the actual question. What is the purpose of the phrase: “It will not affect your farm’s breeding qualifications and will not affect your reputation”? Do you mean that financial penalty is not going to happen to you or your choice will not affect your farm? If the latter, the position of the phrase is not correct. Otherwise you might want to say that the questionnaire is fully anonymous.

6.       The question regarding the detention choices looks also odd. You might want to rephrase it.

7.       It would be interesting for you to report the percentage of farmers who chose the dominated choice in each set (i.e. second case in 10th row). This will give readers a hint about the percentage of respondents that truly understood the meaning of the experimental questions.

8.       In tables 2,3,4 use the same currency unit. Yuan or bucks? State the currency unit used in your experiment.

9.       In table 5, readers can see the farmers’ evaluation regarding punishment policies in general. What about their evaluation regarding financial fines and police detention in particular? Do you have collected data for this?

10.   Rephrase lines 409-413. Your model indicates that the marital status affects the farmers’ evaluation on the necessity of the police detention policy. Not the financial penalty which affects the family income mentioned in line 413.

11.   Please check again the concluding section. Some parts don’t make sense, e.g. lines 464- 465, lines 500-509. In line 473, do you talk about abortion coefficient?

Author Response

Response to Reviewer 2

Comments and Suggestions for Authors

This article investigates how farmers evaluate financial penalties and police detention for the irregular treatment of dead hogs, as well as how specific factors influence their evaluation. The evaluation of punishment policies is divided into three different pillars: the necessity, the degree of implementation and the effectiveness. In addition, based on an economic experiment, the manuscript has the potential to add to the literature on how these components of policy evaluation interact with farmer’s loss aversion. Results indicate that farmers have a positive evaluation on the necessity degree of punishment policies even though they believe that policies are not executed well. Furthermore, loss aversion has some effect on the above mentioned pillars and farmers are more averse to detention than financial penalties. Below follows a list of comments and suggestions for the authors on how the manuscript could be improved.

Broad comments

At some parts of the manuscript, it is extremely difficult for the reader to follow the meaning because of the incoherent English. There are many grammatical incorrect sentences and it would be advisable for the authors to rewrite some parts or take the help of professional editing.

I would suggest the authors to reconsider the title of the paper and use a more declarative and informative title that doesn’t exaggerate the findings.

In introduction and literature review sections, the relevance of the research is not well described, the key hypothesis is difficult to be deducted from these parts and should be elaborated on.

Please re-evaluate the use of the phrase “among them”. It is used too often and sometimes it is used in positions that is not necessary (e.g. lines 107, 354). 

Response.

In response to your suggestion, we have rewrited some parts or take the help of professional editing.

We have now revised the title of the paper as: “Study on the Policy Evaluation of Irregular Treatment of Dead Hogs from the Perspective of Environmental Concern”.

The introduction and literature review sections have been elaborated on.

In introduction and literature review sections, all the variables involved in the text are described. We did not directly make assumptions, but for the key variable loss aversion, we gave literature support.

The use of the phrase “among them” has been re-evaluated.

Specific comments

Point 1. The second paragraph in introduction confuses the reader. In lines 80-83, be more clear on explaining the loss aversion coefficient. In line 84, you might want to talk about “earnings” rather than “income”.

Response 1.

In lines 80-83, we have now rewrote to explain the loss aversion coefficient.

In line 84, we have now rewrote “income” into “earnings”.

Point 2. In line 220, did you actually collect data for aquaculture production? That mean fishes… You use the same word in lines 118 and 486. Please check the meaning.

Response 2.

In line 220, we have now rewrote “aquaculture production condition” into “livestock status”.

In lines 118 and 486, we have now rewrote “aquaculture” into “breeding”.

Point 3. Please elaborate more on the variables description section. In lines 248-249, you describe the variable “culture scale”. What is this? Do you mean the farm scale? And the counting 0-50, 50-500… do you refer to the number of pigs per livestock farm? Please explain. Furthermore, the scale here is not correct. It should be 0-50, 51-500, 501- 3000 etc, otherwise you have overlaps in your calculations.

Response 3.

In line 248-249, we have now rewrote “culture scale” into “farm scale”.

The counting 0-50, 50-500…refer to the number of pigs per livestock farm.Please refer to line 273-274.

We have corrected the scale.Please refer to the line 262.

Point 4. In lines 251-252, please explain why the percentage of the cities below the second line is so bigger than the percentage of the second tier cities and above? Does this disproportion affect your results and how? Please elaborate more on this.

Response 4. This phenomenon is in line with the actual situation. At present, China's second-tier cities and above are banned from raising live pigs. As long as the scope of investigation covers the whole country, the final investigation results will not be affected.

Point 5. In the loss aversion question, there is a discrepancy between the training question and the actual question. What is the purpose of the phrase: “It will not affect your farm’s breeding qualifications and will not affect your reputation”? Do you mean that financial penalty is not going to happen to you or your choice will not affect your farm? If the latter, the position of the phrase is not correct. Otherwise you might want to say that the questionnaire is fully anonymous.

Response 5. In response to your suggestion, we have changed the position of the phrase “It will not affect your farm’s breeding qualifications and will not affect your reputation”.

Point 6. The question regarding the detention choices looks also odd. You might want to rephrase it.

Response 6. In response to your suggestion, we have changed the The question regarding the detention choices.

Point 7. It would be interesting for you to report the percentage of farmers who chose the dominated choice in each set (i.e. second case in 10th row). This will give readers a hint about the percentage of respondents that truly understood the meaning of the experimental questions.

Response 7.We have report the percentage of farmers who chose the dominated choice in each set (i.e. second case in 10th row) in figure 3.

Point 8. In tables 2,3,4 use the same currency unit. Yuan or bucks? State the currency unit used in your experiment.

Response 8. In response to your suggestion, we have now changed bucks into yuan.

Point 9. In table 5, readers can see the farmers’ evaluation regarding punishment policies in general. What about their evaluation regarding financial fines and police detention in particular? Do you have collected data for this?

Response 9. We haven’t collected data about their evaluation regarding financial fines and police detention in particular. We just want to get an evaluation of the overall policy, but the evaluation of each part will become our future research direction.

Point 10. Rephrase lines 409-413. Your model indicates that the marital status affects the farmers’ evaluation on the necessity of the police detention policy. Not the financial penalty which affects the family income mentioned in line 413.

Response 10. We have rephrase lines 409-413.Please refer to 464-466.

Point 11. Please check again the concluding section. Some parts don’t make sense, e.g. lines 464- 465, lines 500-509. In line 473, do you talk about abortion coefficient?

Response 11.

In response to your suggestion, we have deleted lines 464- 465.

In lines 500-509, we made policy recommendations based on conclusions about loss aversion.

In line 473, we have rewrote the sentences.

Reviewer 3 Report

This paper investigates how producers’ degrees of loss aversion are associated with their evaluations about the punishment policies to dead hogs in China by using unique survey data (404 farmers). 

While the survey and research questions are interesting, the analytical framework and interpretations must be improved substantially. To present and interpret the results from the ordered probit models more properly, I recommend the authors to check “Introductory Econometrics” by Wooldridge for the first point, and then “Regression Models for Categorical and Limited Dependent Variables” by Long next.

English writing needs to be improved substantially as well. It is not only about grammars. It is mostly about word choices, structures, and organizations. There is also too much overlapping across sentences and paragraphs (i.e., the same or very similar information are repeated presented). Also, some information seems redundant to the author’s research questions. 

More specific comments are following. 

Major Comments:

I feel a gap between the title and the contents. Although the title includes “sustainable environmental development”, how is “the punishment policy to dead hogs” related to “environmental development”? Also, I found no direct measure of the environment in the paper’s empirical analysis.

Lines 367-375: It is very difficult to understand the explanations about Models I -VI. For example, “The Model І is added to the loss aversion coefficient of the penalty”. This sentence literary means that you add Model I to the loss aversion coefficient. I guess the authors meant to say “loss aversion coefficients are added to Model I”. Also, the name “loss aversion coefficient” is confusing because Table 6 presents the coefficient estimate on the loss aversion coefficient. It may be better to use another name without including “coefficient” in a variable name, for example, “a loss aversion measure”.

In Table 6, because the dependent variables are different across the models (i.e., I&II, II&IV, and V&VI), it should be explicitly presented in the Table. Why don’t you present like “Dependent variable (DV) = PN”, “DV = PI”, and “DV = PE” above every two models? 

Similarly, the following sentences are not clear: “The Model І and Model II are based on the evaluation of the necessary degree of the punishment policy” and similar statements about Models III-VI. Do you mean “the dependent variable in Models I and II is the evaluation of the necessary degree of the punishment policy”, though it is still difficult to understand the meaning of “the evaluation of the necessary degree of”. 

If the results in Table 6 are “coefficients”, you must also present their partial effects to interpret the estimation results of the ordered probit models. In ordered-probit models, even the sign of partial effects can be different from the sign of coefficient estimates. Thus, for example, even when the coefficient estimate on labor is positive, you cannot automatically claim a positive effect on the dependent variables. The sign of the effect (i.e., a partial effect) depends on the values of labor and all other explanatory variables. In addition, you should interpret not only the sign but also the magnitude of the effect. The authors should be able to estimate the effect on the probability of choosing each category (e.g., the effect on the probability of choosing “Unnecessary = 2” in the necessary for the punishment policy), separately. The same comment is applied to the interpretation of all other variables. 

It will be much better if the associations with loss aversions (i.e., Tanaka’s measures) is compared with the associations with risk aversions (i.e., a conventional risk aversion measure). Though this depends on whether the authors collected data on risk aversions, the evaluations about punishment policies are also very likely to be correlated with risk aversions. Theoretically, there is no clear reason to believe that loss aversion is a dominant factor to explain the evaluation compared to risk aversion. This point is rather an empirical question, and thus it should be examined empirically. 

In conclusions, the authors claim as if all their explanatory variables ‘affect’ the policy evaluations. But, it will be safer to claim only ‘associations’ because some explanatory variables seem simultaneously determined with the evaluations. For example, both breeding period and the evaluations may be the outcomes that are determined by another more fundamental factor (e.g., more careful people may increase breeding period and positively (or negatively) evaluate the policy), rather than that breeding period determines the evaluations. 

Similarly, the following claim is too strong: “For the problem of endogeneity, this paper selects as many control variables as possible that affect both the explanatory variables and the explained variables, so that endogenous problems such as missing variables can be solved. ” This is true only when you include “exogenous” explanatory variables. If you include endogenous variables as controls, this statement does not hold. Also, even if you solve the problem of missing variables, I still see there remain simultaneity problem which also causes endogeneity. That is, the dependent variables and some explanatory variables do not have a clear time order in its decision process, and thus they may be simultaneously determined without a clear causal relationship.  

About policy implications, while the authors listed many policy implications, they should focus on the implications that are newly derived from their results. Currently, most implications are trivial and not directly implied from their results. Because the key contribution of this paper is estimating the association between loss aversion and the policy evaluations, the authors should focus on the implications directly derived from the findings. 

Minor comments:

In equation (1), loss aversion measures are LA1 and LA2. But, in Table 6, there are only Risk 1 and Risk 2. Aren’t they the same variables? 

In Table 5, is there any difference across farm sizes? For example, larger farmers may be more positively evaluate the policies compared to smaller farmers, or vice versa. If so, it may be better to present the summary statistics separately by farm size.

In Table 6, you can simply say “coefficient” instead of “Estimation coefficient”. 

After Table 6, when you interpret the estimation results, you should start from the key variables for your research questions. I believe your key variables are loss aversion coefficients. So, you should start interpreting the result for loss aversion.

Author Response

Response to Reviewer 3

Comments and Suggestions for Authors

This paper investigates how producers’ degrees of loss aversion are associated with their evaluations about the punishment policies to dead hogs in China by using unique survey data (404 farmers). 

While the survey and research questions are interesting, the analytical framework and interpretations must be improved substantially. To present and interpret the results from the ordered probit models more properly, I recommend the authors to check “Introductory Econometrics” by Wooldridge for the first point, and then “Regression Models for Categorical and Limited Dependent Variables” by Long next.

English writing needs to be improved substantially as well. It is not only about grammars. It is mostly about word choices, structures, and organizations. There is also too much overlapping across sentences and paragraphs (i.e., the same or very similar information are repeated presented). Also, some information seems redundant to the author’s research questions. 

More specific comments are following. 

Major Comments:

Point 1. I feel a gap between the title and the contents. Although the title includes “sustainable environmental development”, how is “the punishment policy to dead hogs” related to “environmental development”? Also, I found no direct measure of the environment in the paper’s empirical analysis.

Response 1. In response to your suggestion, we have now revised the title of the paper as: “Study on the Policy Evaluation of Irregular Treatment of Dead Hogs from the Perspective of Environmental Concern”,

Point 2. Lines 367-375: It is very difficult to understand the explanations about Models I -VI. For example, “The Model І is added to the loss aversion coefficient of the penalty”. This sentence literary means that you add Model I to the loss aversion coefficient. I guess the authors meant to say “loss aversion coefficients are added to Model I”. Also, the name “loss aversion coefficient” is confusing because Table 6 presents the coefficient estimate on the loss aversion coefficient. It may be better to use another name without including “coefficient” in a variable name, for example, “a loss aversion measure”.

Response 2. We have rewrote the explanations about Models I -VI. Please refer to line 421-432.We also changed “loss aversion coefficient” into “a loss aversion measure”.

Point 3. In Table 6, because the dependent variables are different across the models (i.e., I&II, II&IV, and V&VI), it should be explicitly presented in the Table. Why don’t you present like “Dependent variable (DV) = PN”, “DV = PI”, and “DV = PE” above every two models? 

Response 3. We also added “DV = PN”, “DV = PI”, and “DV = PE” in table 6.

Point 4. Similarly, the following sentences are not clear: “The Model І and Model II are based on the evaluation of the necessary degree of the punishment policy” and similar statements about Models III-VI. Do you mean “the dependent variable in Models I and II is the evaluation of the necessary degree of the punishment policy”, though it is still difficult to understand the meaning of “the evaluation of the necessary degree of”. 

Response 4. We also changed“The Model І and Model II are based on the evaluation of the necessary degree of the punishment policy”into “the dependent variable in Models I and II is the evaluation of the necessary degree of the punishment policy”, and similar statements about Models III-VI.

The evaluation of the necessary degree of policy refers to the evaluation of whether the policy itself is worthy of existence, and is one of the indicators of evaluation.

Point 5. If the results in Table 6 are “coefficients”, you must also present their partial effects to interpret the estimation results of the ordered probit models. In ordered-probit models, even the sign of partial effects can be different from the sign of coefficient estimates. Thus, for example, even when the coefficient estimate on labor is positive, you cannot automatically claim a positive effect on the dependent variables. The sign of the effect (i.e., a partial effect) depends on the values of labor and all other explanatory variables. In addition, you should interpret not only the sign but also the magnitude of the effect. The authors should be able to estimate the effect on the probability of choosing each category (e.g., the effect on the probability of choosing “Unnecessary = 2” in the necessary for the punishment policy), separately. The same comment is applied to the interpretation of all other variables. 

Response 5.In response to your suggestion, we added partial effects to interpret the estimation results of the ordered probit models, and we changed the interpretation.

Point 6. It will be much better if the associations with loss aversions (i.e., Tanaka’s measures) is compared with the associations with risk aversions (i.e., a conventional risk aversion measure). Though this depends on whether the authors collected data on risk aversions, the evaluations about punishment policies are also very likely to be correlated with risk aversions. Theoretically, there is no clear reason to believe that loss aversion is a dominant factor to explain the evaluation compared to risk aversion. This point is rather an empirical question, and thus it should be examined empirically. 

Response 6. First of all, we did not collect data on risk aversions. The reason we use loss aversion rather than risk aversion is that the punishment policy causes people to lose property or personal freedom.

Point 7. In conclusions, the authors claim as if all their explanatory variables ‘affect’ the policy evaluations. But, it will be safer to claim only ‘associations’ because some explanatory variables seem simultaneously determined with the evaluations. For example, both breeding period and the evaluations may be the outcomes that are determined by another more fundamental factor (e.g., more careful people may increase breeding period and positively (or negatively) evaluate the policy), rather than that breeding period determines the evaluations. 

Response 7. First of all, we didn’t claim all explanatory variables ‘affect’ the policy evaluations. Some variables don’t have Influence.

And, In response to your suggestion, we have now revised ‘affect’ to ‘associations’.

Point 8. Similarly, the following claim is too strong: “For the problem of endogeneity, this paper selects as many control variables as possible that affect both the explanatory variables and the explained variables, so that endogenous problems such as missing variables can be solved. ” This is true only when you include “exogenous” explanatory variables. If you include endogenous variables as controls, this statement does not hold. Also, even if you solve the problem of missing variables, I still see there remain simultaneity problem which also causes endogeneity. That is, the dependent variables and some explanatory variables do not have a clear time order in its decision process, and thus they may be simultaneously determined without a clear causal relationship.  

Response 8. In response to your suggestion, we have weakened our claims as “For the problem of endogeneity, this paper selects as many control variables as possible that affect both the explanatory variables and the explained variables, so that endogenous problems such as missing variables can be solved almost.” For the description of simultaneity problem, we have seen some examples of enough control variables to solve such problems, so we give such a claim.

Point 9. About policy implications, while the authors listed many policy implications, they should focus on the implications that are newly derived from their results. Currently, most implications are trivial and not directly implied from their results. Because the key contribution of this paper is estimating the association between loss aversion and the policy evaluations, the authors should focus on the implications directly derived from the findings. 

Response 9.As the suggestion say, our key contribution of this paper is estimating the association between loss aversion and the policy evaluations , in the policy recommendations, recommendations 3-6 focus on the implications directly derived from the findings. 

Minor comments:

Point 1. In equation (1), loss aversion measures are LA1 and LA2. But, in Table 6, there are only Risk 1 and Risk 2. Aren’t they the same variables? 

Response 1. In response to your suggestion, we have changed the equation (1).

Point 2. In Table 5, is there any difference across farm sizes? For example, larger farmers may be more positively evaluate the policies compared to smaller farmers, or vice versa. If so, it may be better to present the summary statistics separately by farm size.

Response 2. We have already used size as an influencing factor, so we don’t need to go into details.

Point 3. In Table 6, you can simply say “coefficient” instead of “Estimation coefficient”. 

Response 3. In response to your suggestion, we have changed “Estimation coefficient” into “coefficient”.

Point 4. After Table 6, when you interpret the estimation results, you should start from the key variables for your research questions. I believe your key variables are loss aversion coefficients. So, you should start interpreting the result for loss aversion.

Response 4. In response to your suggestion, we have changed the order of interpretation.

Round 2

Reviewer 1 Report

The authors did make major revision from the last version. The manuscript has been revised, while still, there are some small issues need to be solved before publication.

1.      The research methods used in the paper should be present in separate part, not in the empirical analysis. Section 3 is Theoretical Framework, Section 4 is Methods and Data. Section 5 is Results and discussions. Please put the relevant content in the corresponding section. Please reorganize your research structure according to other scientific papers.

2.      Please avoid using "We" in writing the scientific paper.

3.      Please add the legend in the Fig.1, rather than notes.

4.      Fig3, Titles for vertical and horizontal axes should be placed in appropriate positions.

5.      Please revise and improve all the graphs of the article according to other academic papers.

Author Response

Reviewer 1

The authors did make major revision from the last version. The manuscript has been revised, while still, there are some small issues need to be solved before publication.

Point 1. The research methods used in the paper should be present in separate part, not in the

empirical analysis. Section 3 is Theoretical Framework, Section 4 is Methods and Data. Section 5 is Results and discussions. Please put the relevant content in the corresponding section. Please reorganize your research structure according to other scientific papers.

Response 1. Thank you for your valuable comments. In response to your suggestion, we have changed the order of the parts of the article.

Point 2. Please avoid using "We" in writing the scientific paper.

Response 2. Thank you for your valuable comments. In response to your suggestion, we have changed “we” into “this paper”

Point 3. Please add the legend in the Fig.1, rather than notes.

Response 3. Thank you for your valuable comments. In response to your suggestion, we have changed the notes into legend.

Point 4. Fig3, Titles for vertical and horizontal axes should be placed in appropriate positions.

Response 4. Thank you for your valuable comments. In response to your suggestion, we have changed the position of the titles for vertical and horizontal axes.

Point 5. Please revise and improve all the graphs of the article according to other academic papers.

Response 5. Thank you for your valuable comments. In response to your suggestion, we have changed all the graphs of the article according to other academic papers.

Reviewer 3 Report

The revisions for Point 5 is not sufficient at all. The authors have to interpret what Margin (1) to Margin (5) are. The authors just listed the number and hardly interpret the number in a meaningful way. The authors must learn ordered logit model more.  

Responses 7 and 8 are contradicting. If you claim like in response 8, you do not need to say "associations". The authors should choose one side, and I think the endogeneity problem is not solved at all in this paper.   

Why do the authors want to include "environment"? I still do not get which part of the analysis is related to environmental protection. For example, in Table 1, the authors classify "disposal" and "cooperative" as external environment. If the variables are so important, the authors should explain them in more detail. I could not find any clear definition of the variables. What exactly means by disposal = 1 or cooperative=1 in reality?  And, are they really "environment" in what sense? I suspect the authors talking about institutions or market environments.   

Author Response

Reviewer 3:

Point 1.The revisions for Point 5 is not sufficient at all. The authors have to interpret what Margin (1) to Margin (5) are. The authors just listed the number and hardly interpret the number in a meaningful way. The authors must learn ordered logit model more.

Response 1. Thank you for your valuable comments. In response to your suggestion, we have give the interpretation of Margin (1) to Margin (5) in line 405~407. We have gave interpretation the number in a meaningful way in line 426~504.

Point 2.Responses 7 and 8 are contradicting. If you claim like in response 8, you do not need to say

"associations". The authors should choose one side, and I think the endogeneity problem is not solved at all in this paper.

Response 2. Thank you for your valuable comments. We choose the “affect” instead of “associations”. Because I read some of the methods in the literature (Schijven and Hitt, 2012; Foss , Frederiksen and Rullani, 2016) that address endogenous problems, control variables are used to solve the problem.

Point 3.Why do the authors want to include "environment"? I still do not get which part of the analysis is related to environmental protection. For example, in Table 1, the authors classify "disposal" and "cooperative" as external environment. If the variables are so important, the authors should explain them in more detail. I could not find any clear definition of the variables. What exactly means by disposal = 1 or cooperative=1 in reality? And, are they really "environment" in what sense? I suspect the authors talking about institutions or market environments.

Response 3. Thank you for your valuable comments. In response to your suggestion, we have now revised the title of the paper as: “Evaluation of Policies on Inappropriate Treatment of Dead Hogs from the Perspective of Environmental Concerns Loss Aversion”. We deleted the "environment concern" in the list of key words. We clear the "disposal" and "cooperative" variables in the line 231~233. “Disposal = 1” means that there are dead pig treatment points in the breeding area. “Cooperative=1” means that farmers participate in pig breeding cooperatives. The sentence “What needs special explanation is that “External environmental characteristics” mean the support and help that the government and society can provide during the breeding process” can explain the word “environment”.